# Stable and Effective Trainable Decoding for Sequence to Sequence Learning

**Yun Chen**[†]**, Kyunghyun Cho**[‡]**, Samuel R. Bowman,** [‡] **Victor O.K. Li**[†]
[†]The University of Hong Kong, [‡]New York University
`yun.chencreek@gmail.com, kyunghyun.cho@nyu.edu`
`bowman@nyu.edu, vli@eee.hku.hk`

## Abstract

We introduce a fast, general method to manipulate the behavior of the decoder in a sequence to sequence neural network model. We propose a small neural network *actor* that observes and manipulates the hidden state of a previously-trained decoder. We evaluate our model on the task of neural machine translation. In this task, we use beam search to decode sentences from the plain decoder for each training set input, rank them by BLEU score, and train the actor to encourage the decoder to generate the highest-BLEU output in a single *greedy* decoding operation without beam search. Experiments on several datasets and models show that our method yields substantial improvements in both translation quality and translation speed over its base system, with no additional data.

## 1 Introduction

Neural network sequence decoders have yielded state-of-the-art results for many text generation tasks such as machine translation (Bahdanau et al., 2015; Luong et al., 2015; Gehring et al., 2017; Vaswani et al., 2017), text summarization (Rush et al., 2015; Ranzato et al., 2015) and image captioning (Vinyals et al., 2015; Xu et al., 2015). They generate words from left to right, at each step giving a distribution over possible next words conditioned over all words generated so far. However, since the space of possible output sequences is exponentially large, heuristic search methods such as greedy decoding or beam search (Graves, 2012; Boulanger-Lewandowski et al., 2013) must be used to find high-probability sequences. Unlike greedy decoding which selects the word of the highest probability at each step, beam search expands all possible next steps and keeps the $B$ most likely ones, where $B$ is the beam size. Greedy decoding is very fast—requiring only a single run of the underlying decoder—while beam search requires several such passes, with substantial additional overhead for data management, but often leads to substantial improvement over greedy decoding. For example, Ranzato et al. (2015) report that beam search (with a width of 10) gives a 2.2 BLEU improvement in translation and a 3.5 ROUGE-2 improvement in summarization over greedy decoding.

In this paper, we propose a procedure to modify a trained decoder to allow it to generate text greedily with the same quality level that would otherwise be seen with the much slower (small-beam) beam search procedure. To do this, we introduce a small neural network *actor*, which observes and manipulates the hidden state of the decoder. Previously, Gu et al. (2017) try to train a similar actor with critic-aware actor learning algorithm. However, due to the noisy gradient estimation of the critic, their model is able to train only with carefully designed training method and actor architecture. Thus, we propose to train this actor with an artificial parallel corpus generated by beam search on the pre-trained sequence-to-sequence model with beam $B = 35$ such that greedy decoding with the actor finds a sequence of higher quality. Our method is stable to train, effective, and can be easily employed with a variety of decoder architectures. We demonstrate this for neural machine translation on three state-of-the-art architectures and two corpora.

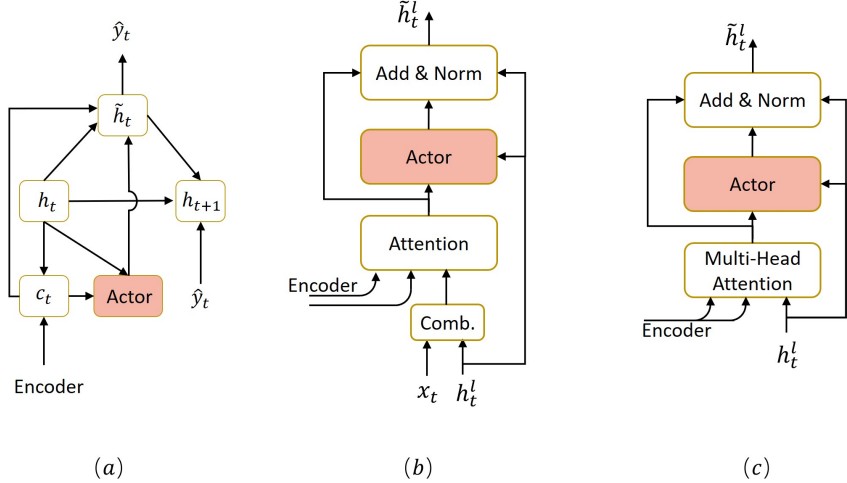

$$(a) \qquad\qquad\qquad (b) \qquad\qquad\qquad (c)$$

Figure 1: A single step of the actor interacting with the underlying neural translation model. (a) RNN-based NMT model (Luong et al., 2015); (b) ConvS2S (Gehring et al., 2017); (c) Transformer (Vaswani et al., 2017).

## 2 METHODS

**Architecture** As in Gu et al. (2017), we introduce a small *actor* network to manipulate the state of the decoder in the underlying sequence to sequence model. This actor takes as input the current decoder state, the source context vector (output of the attention network) and produces a vector-valued action which is added back to the hidden state. We implement the actor as a small gated feedforward neural network.

Below, we use $h$ to denote the current decoder state and $c$ to denote the source context vector, and define the action $a$ as

$$a = z \circ Wh + (1-z) \circ Uc, \tag{1}$$

where

$$z = \sigma(W_z h + U_z c). \tag{2}$$

This gate is similar to the context gate (Tu et al., 2017). The actor can decide whether to rely more on the source context $c$ or on the decoder state $h$ to generate the action. After getting the action $a$, the hidden state $h$ can be updated as:

$$\tilde{h} = f(h, c) + a \tag{3}$$

Fig. 1 shows the a single step of the actor interacting with the underlying neural translation model for different NMT architectures. From left to right, it represents RNN-based NMT model (Luong et al., 2015), ConvS2S (Gehring et al., 2017) and Transformer (Vaswani et al., 2017), respectively. For RNN-based NMT, we add the action $a_t$ to the attention vector $\tilde{h}_t$. For ConvS2S and Transformer, we add the action $a_i^l$ to the decoder state $\tilde{h}_i^l$.

**Learning** To overcome the instability problem of Gu et al. (2017), we propose to train the actor with a pseudo-parallel corpus generated from the underlying NMT model (Kim & Rush, 2016; Freitag et al., 2017; Zhang et al., 2017). We want a corpus that trades off the two goals of (i) having a high model likelihood, so we can coerce the model to generate it without too much additional training or too many new parameters and, (ii) having a good translation quality, measured by BLEU. We do this by generating sentences from the original model with beam search ($B = 35$), and choosing the generated sentence with the highest BLEU for each input.

More specifically, let $\langle \mathbf{x}, \mathbf{y} \rangle$ be a sentence pair in the training data and $\mathbb{Z} = \{\mathbf{z}^1, ..., \mathbf{z}^B\}$ be the beam search results from the pretrained NMT model, where $B$ is the beam size. We measure the smoothed sentence-level BLEU score (Papineni et al., 2002) between the model prediction $\mathbf{z}$ and the

|  |  | greedy | tok/sec | beam4 | tok/sec | greedy | tok/sec | beam4 | tok/sec |
|---|---|---|---|---|---|---|---|---|---|
| IWSLT16 |  | De → En | | | | En → De | | | |
| RNN-based | base | 23.57 | 52.4 | 24.90 | 39.6 | 20.05 | 39.5 | 21.11 | 27.4 |
|  | actor | 23.77 | 50.2 | 25.18 | 37.1 | 19.98 | 34.7 | 20.74 | 25.3 |
| ConvS2S | base | 27.44 | 152.3 | 28.80 | 80.5 | 22.88 | 122.4 | 24.02 | 58.5 |
|  | actor | 28.80 | 136.7 | **29.39** | 72.8 | 24.32 | 103.8 | 24.80 | 52.9 |
| Transformer | base | 27.15 | 61.7 | 28.74 | 29.9 | 23.87 | 53.7 | 25.03 | 25.4 |
|  | actor | 28.43 | 55.3 | 29.18 | 28.1 | 25.65 | 48.3 | **25.68** | 23.7 |
| WMT15 |  | Fi → En | | | | En → Fi | | | |
| RNN-based | base | 12.45 | 41.9 | 13.22 | 29.6 | 9.77 | 43.3 | 10.81 | 30.8 |
|  | actor | 13.07 | 37.8 | 13.50 | 28.5 | 10.37 | 41.7 | 11.04 | 29.9 |
| ConvS2S | base | 15.43 | 24.3 | 16.86 | 11.4 | 12.65 | 23.3 | 13.97 | 10.9 |
|  | actor | 17.17 | 16.2 | **17.59** | 7.4 | 14.42 | 15.5 | **14.79** | 7.1 |
| Transformer | base | 13.76 | 29.2 | 14.61 | 12.4 | 12.38 | 28.6 | 13.55 | 12.2 |
|  | actor | 14.26 | 26.6 | 14.88 | 11.8 | 12.91 | 25.8 | 13.46 | 11.4 |

Table 1: Generation quality (BLEU↑) and speed (tokens/sec↑). Speed is measured assuming sentence-by-sentence generation without mini-batching on the test set with CPU.

gold-standard translation $\mathbf{y}$. Then we choose the sentence $\tilde{\mathbf{z}}$ that has the highest BLEU score as our new target sentence:

$$\tilde{\mathbf{z}} = \underset{\mathbf{z}^b = \mathbf{z}^1, \ldots, \mathbf{z}^B}{\operatorname{argmax}} \operatorname{BLEU}(\mathbf{z}^b, \mathbf{y}). \tag{4}$$

Once we obtain the pseudo corpus $D_{x,z}$, we keep the underlying model fixed and train the actor by maximizing the log-likelihood of the pseudo pairs:

$$\hat{\boldsymbol{\theta}}_{actor} = \underset{\boldsymbol{\theta}_{actor}}{\operatorname{argmax}} \left\{ \sum_{\langle \mathbf{x}, \mathbf{z} \rangle \in D_{x,z}} \log P(\mathbf{z}|\mathbf{x}; \hat{\boldsymbol{\theta}}_{nmt}, \boldsymbol{\theta}_{actor}) \right\}. \tag{5}$$

## 3 EXPERIMENTS

**Datasets** In this paper, we evaluate the proposed model on IWSLT16 English-German and WMT15 English-Finnish translations on both directions over three SOTA translation architectures: RNN-based NMT, ConvS2S and Transformer. For IWSLT16, we use tst2013 and tst2014 for validation and testing, respectively. For WMT15, we use newstest2013 and newstest2015 for validation and testing, respectively. All the data are tokenized and segmented into subword symbols using byte-pair encoding (BPE) (Sennrich et al., 2016) to restrict the size of the vocabulary. We evaluate using tokenized and cased BLEU scores (Papineni et al., 2002).

**Results and Analysis** The results, in Table 1, show that the use of our actor makes it practical to replace beam search with greedy search in most cases: we lose little or no performance, and doing so yields a dramatic increase in speed, even accounting for the small overhead from the actor. In cases where translation quality is much more valuable than raw throughput, our method can also be combined with beam search at test time to yield results somewhat better than either could achieve alone. Among these three translation models, ConvS2S and Transformer benefit more from the actor, with 1.58 BLEU and 1.02 BLEU improvement on average for greedy decoding when using the actor.

## 4 CONCLUSION

In this paper, we propose a simple method that adds few parameters, but allows the sequence to sequence model to decode with less time and memory. Our extensive experiments on three translation systems and two parallel corpora confirmed its validity and usefulness.

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
