# OpenReview forum: "Stable and Effective Trainable Greedy Decoding for Sequence to Sequence Learning"
_ICLR.cc/2018/Workshop — Accept_

### Official Review · AnonReviewer3 · 2018-03-06
**Light-weight extension of Gu et al. (2017) with Kim and Rush (2016)**

**Rating:** 6
**Confidence:** 4

**Review:**

The paper addresses instability issues in Gu et al. (2017) by training a differentiable actor network on a pseudo-parallel corpus generated by a beam model (Kim and Rush, 2016): this can be viewed as beam-to-greedy knowledge distillation. Major NMT models (RNN, ConvS2S, Transformer) are all shown to benefit from this technique. The approach is a bit thin in its technical novelty and the benefit is marginal in many cases. However, perhaps the technique is simple enough to be useful in practice.

---

### Official Review · AnonReviewer2 · 2018-03-08
**On a new effective trainable greedy decoding**

**Rating:** 6
**Confidence:** 1

**Review:**

First I need to admit that this paper is very far from my experience. I am briefly understanding that the author is trying to introduce a new actor network to manipulate the state of the decoder. As far as I see, the numerical experiments look fairly good.

---

### Official Review · AnonReviewer1 · 2018-03-13
**using ``````````"actor" network to achieve comparable performance in S2S tasks with greedy decoding than compared with beam search scheme**

**Rating:** 6
**Confidence:** 2

**Review:**

This paper proposes a method to manipulate decoders in sequence to sequence learning: a small neural network
actor is used to update hidden state of a previously-trained decoder. The authors evaluate the model on the task of neural machine translation.

The motivation of this work is to combat the computational inefficiency of beam search in manipulating the decoder, such that even with the greedy scheme (which is fast) one can still preserve the performance. However, this method requires using a beam search to decode sentences from the plain decoder for the training set, and then training the actor to encourage decoder to generate the highest-BLEU output in a single greedy decoding operation. However, this makes the training more costly, and also, why not let the actor learn directly from the ground truth rather than from the beam-search-generated results? Is there a specific advantage with it?

The authors should also spend more effort describing the results in Table~1, for example, what is meant by bold numbers, and the underlined numbers?

---

### Decision · Program_Chairs · 2018-03-20
**ICLR 2018 Workshop Acceptance Decision**

**Decision:**

Accept

**Comment:**

Congratulations, your paper was accepted to the ICLR workshop.